# Comparison of Thermal and High-Pressure Pasteurization on Immunoglobulins, Lysozyme and Microbial Quality of Donkey Colostrum

Mafalda S. Gonçalves [1], Liliana G. Fidalgo [1,2,3], Silvia G. Sousa [1], Rui P. Queirós [1,4], Sónia M. Castro [1], Carlos A. Pinto [1] and Jorge A. Saraiva [1,*]

1   LAQV-REQUIMTE, Department of Chemistry, University of Aveiro, 3810-193 Aveiro, Portugal; mafalda.sofia87@gmail.com (M.S.G.); liliana.fidalgo@ipbeja.pt (L.G.F.); mariasgomessousa@gmail.com (S.G.S.); r.queiros@hiperbaric.com (R.P.Q.); smcastro@ua.pt (S.M.C.); carlospinto@ua.pt (C.A.P.)
2   Department of Technology and Applied Sciences, School of Agriculture, Polytechnic Institute of Beja, 7800-309 Beja, Portugal
3   MED—Instituto Mediterrâneo para a Agricultura, Ambiente e Desenvolvimento & CHANGE–Global Change and Sustainability Institute, CEBAL, 7800-309 Beja, Portugal
4   Hiperbaric SA, Poligono Industrial Villalonquejar, Calle Condado de Treviño, 6, 09001 Burgos, Spain
*   Correspondence: jorgesaraiva@ua.pt

**Abstract:** The effect of thermal pasteurization (TP, 62.5 °C/30 min—conditions similar to those used in milk banks/hospitals, known as Holder pasteurization) and high-pressure pasteurization (HPP: 400–625 MPa/2.5–30 min) was studied on immunoglobulin (IgG, IgM and IgA) content, lysozyme activity and microbial load of donkey colostrum (in this case, after 40 days at 4 °C). IgG level remained unchanged with HPP at 400 MPa, increased up to 4-fold at 625 MPa/10 min and decreased 90% with TP, while IgM decreased progressively with pressure treatment intensity increment to below the detection limit at 625 MPa and decreased 20% with TP. IgA decreased to below the detection limit after TP and all HPP treatments. Lysozyme activity presented overall a higher decrease after TP (37%) compared to HPP (decreasing from 20% at 400 MPa to about 40% at 600 MPa/10 and 30 min). Furthermore, both total aerobic mesophiles and Enterobacteriaceae remained below detection limits after 40 days of refrigerated storage for both TP and HPP. So, HPP can be considered a potential alternative to the conventional TP to preserve donkey colostrum, with overall equal to better retention, particularly for IgG and lysozyme activity. As far as the authors are aware, this is the first study evaluating the effects of HPP on donkey colostrum, and research in this field should be pursued.

**Keywords:** donkey colostrum; high-pressure; thermal pasteurization; microbial load; immunoglobulins; IgG; IgM; IgA

## 1. Introduction

Finding alternatives to human milk is crucial when breastfeeding is impossible and cow milk is unobtainable. According to several authors, donkey milk is the best human milk substitute [1], mainly because its composition is clearly more similar to human milk than that of other animals (cow, buffalo, sheep, goat and camel [2]) and additionally due to its nutrient and functional compound richness [3].

Colostrum, the initial secretion of milk in mammals, is an important fluid that contains bioactive components, namely immunoglobulins (Igs: IgA, IgM and IgG) and substantial concentrations of antimicrobial factors, including elevated levels of various enzymes (e.g., lysozyme) possessing antimicrobial properties [4]. Even though colostrum is richer in dry matter, protein, fat, vitamins and minerals (except calcium and phosphorus), it is poorer in lactose than mature milk [5].

Both human colostrum and milk are currently processed by thermal pasteurization in low-temperature/long-time conditions (62.5 °C, 30 min) in human milk banks, a process known as Holder pasteurization [6,7]. However, it has been verified that when TP is applied to human colostrum, a decrease in IgA, IgM and IgG is observed, for instance, in values around 20, 51 and 23%, respectively [7], thus reducing the nutritional and functional quality.

In lieu of traditional thermal pasteurization (TP), high-pressure pasteurization (HPP) without the application of heat can serve as a nonthermal method for pasteurization. HPP has the capability to inactivate microorganisms, resulting in the production of safe and minimally processed food products. Additionally, HPP helps to preserve the nutritional and organoleptic qualities of the food to a greater extent [8]. No publications were found concerning HPP's effect on the microbiological quality and immunologic parameters (Igs and lysozyme activity) of donkey colostrum, although there are a few studies on donkey milk [9,10] but dealing only with microbial quality.

Giacometti et al. (2016) [10] studied the effects of TP (65 °C, 30 min), HPP (400 MPa, 180 s) and TP+HPP (65 °C, 30 min + 400 MPa, 100 s) on microbiological indicators of donkey milk hygiene and their evolution during storage at 4 and 12 °C for 30 days, simulating a farm-scale pasteurization and packing system. In this study, it was verified that the HPP treatment of pasteurized milk extended the shelf-life of donkey milk and assured its microbial criteria for consumption up to 30 days if properly stored at 4 °C, verifying that combining both TP and HPP treatments allowed the enhancement of donkey milk shelf-life from 3 to 30 days.

Recently, Dambrosio et al. (2023) [9] investigated the viability of *Escherichia coli* O157:H7 in raw donkey milk, as well as in thermally pasteurized (50 °C for 30 min, TP) and HPP (400 MPa at 3 °C for 60 s) milks, and by combining TP+HPP. The authors reported that *E. coli* O157:H7 exhibited viability in untreated donkey milk for a duration of 15 days, but in TP milk, it survived for a period of 4 days. The application of HPP effectively eliminated the pathogen, as it was not found from the beginning of the treatment. Hence, the application of HPP demonstrated efficacy in ensuring the microbiological integrity of the product. Regarding the effect of HPP on immunologic parameters (Igs and lysozyme activity), there are a few studies on colostrum from other species but none on donkey colostrum, for instance in caprine [11] and human [7,12] colostrum. For caprine colostrum, treatment at 400 and 500 MPa caused a decrease in the IgG content from 20 to 40%, respectively [11]. In human colostrum, low pressure levels (200–400 MPa) did not affect Ig content, but contrarily, 600 MPa (2.5 min) promoted IgM and IgG losses (21%) despite the maintenance of IgA. Longer treatments (600 MPa, 15 and 30 min) caused similar or higher Igs losses comparatively to TP [7], while lysozyme seemed to be resistant in HPP-treated human colostrum (statistically similar to unprocessed human colostrum) when compared to TP, which decreased 44%. In another study, Foster et al. (2016) [12] verified that increasing holding time at 400 MPa caused a considerable decline in the IgG concentration in bovine colostrum, but no significant differences were observed at 300 MPa. In another study with bovine colostrum, Masura et al. (2000) [13] reported that HPP at 300 and 400 MPa for 10 min at room temperature effectively suppressed the development of total aerobic mesophiles, coliforms and psychrotrophic Gram-negative bacteria under refrigeration for 9 days. The authors also stated that, at 300 MPa, the IgG of colostrum was minimally affected, as the neutralizing ability against bovine coronavirus was preserved, while 400 MPa resulted in IgG denaturation (observed qualitatively) and changes in colostrum viscosity (observed qualitatively).

In another study, Wesolowska et al. (2018) [14] compared the effects of single and cycled HPP treatments (600 MPa/10 min, 100 MPa/10 min–10 min interval–600 MPa/ 10 min, 200 MPa/10 min–10 min interval–400 MPa/10 min and 200 MPa/10 min–10 min interval–600 MPa/10 min, at 19–21 °C) and traditional Holder pasteurization (62.5 °C, 30 min) in human milk. The results showed that the condition 200 + 400 MPa was able to better preserve the IgG levels compared to traditional Holder pasteurization (82.24 versus 50.96%, respectively), as well as the levels of hepatocyte growth factor (97.15 versus 11.28%,

respectively). The authors also reported that the content of leptin increased for all HPP conditions tested compared to Holder pasteurization, as well as a better retention of insulin.

Other technologies, such as ultraviolet radiation (UV-radiation) have been addressed for colostrum pasteurization. For example, Teixeira et al. (2013) [15] studied the effects of UV-radiation (45 J/cm$^2$ at 285 nm) for nonthermal pasteurization of bovine colostrum, with the results suggesting that this methodology performed poorly compared to conventional thermal processing (63 °C for 30 min and 72 °C for 15 s), mostly due to the low penetrance of UV-radiation due to the colostrum's high level of solids. Truly, it should be highlighted that the existence of dissolved and suspended solids has the capability to disperse UV light and create an environment for bacterial aggregation, with both factors thereby diminishing the bactericidal effectiveness of UV.

Since there are no reports concerning the HPP effect on donkey colostrum, the aim of this work was to study and compare the effect of TP (62.5 °C, 30 min) and HPP (400, 550 and 625 MPa for 2.5, 10 and 30 min, at 8 °C) on the endogenous microbial load, Igs (IgA, IgM and IgG) content and lysozyme activity of donkey colostrum after processing and during subsequent refrigerated storage (4 °C) for 40 days.

## 2. Materials and Methods

### 2.1. Donkey Colostrum Collection and Processing

The colostrum used in this study was generously donated by a local commercial donkey milk farm that produces donkey milk and colostrum to sell and/or to create donkey milk-based products, for example, dried milk and cosmetic products. The colostrum was collected from an apparently healthy female donkey, one day after partum, and transported to the laboratory within four hours in refrigerated conditions (4 °C). At the time of this study, the farm had only one female with available colostrum suitable for our research. The limited availability of colostrum for research purposes was due to its critical role in supporting the health of foals on the farm, and the donation volume was restricted accordingly.

In the laboratory and under aseptic conditions, the colostrum was equally divided for thermal (TP) and high-pressure pasteurization (HPP). For TP, donkey colostrum samples were thermally pasteurized using the Holder method, according to the general HM-BANA procedure [16], with colostrum aliquots (3.0 mL) placed into previously sterilized polypropylene tubes at 62.5 °C for 30 min using a thermostatic water bath. For HPP, donkey colostrum was aseptically added into low permeability polyamide–polyethylene bags (3 mL, PA/PE-90, Plásticos Macar—Indústria de Plásticos Lda., Palmeira, Portugal). HPP treatments were performed by combining three pressure levels and three holding times: 400, 550 and 625 MPa for 2.5, 10 and 30 min using a 100 mL hydrostatic press (Unipress Equipment, Model U33, Warsaw, Poland) surrounded by an external jacket connected to a thermostatic bath to control the temperature, with a mixture of propylene glycol and water (60:40) as the pressure-transmitting fluid. The initial temperature of the pressure vessel was set to 8 °C while the colostrum samples were initially at 4 °C.

After TP and HPP, samples were evaluated for microbial quality within 1 h maximum (day 0 of sampling) and stored at 4 °C for 40 days to study the evolution of microbial quality, whereas, for immunologic parameters, samples were stored at −80 °C until the analyses were carried out. Microbial quality was evaluated in HPP samples treated at 400 and 550 MPa for 10 min during refrigerated storage for 40 days. This range of pressures and times was selected since they fit the current conditions used in the food industry for HPP products.

### 2.2. Microbial Load Analysis

Each sample (1.0 mL) was obtained aseptically and homogenized in 9.0 mL of sterile Ringer's solution. Further, decimal dilutions were made with the same Ringer's solution and duplicates of dilutions were plated on the appropriate media, according to the procedures described below. Total aerobic mesophilic (TAM) counts at 30 °C were determined in plate count agar (PCA, Merck, Burlington, MA, USA) and, after incubation at 30 °C

for 72 h, the colonies formed were then counted [17]. Enterobacteriaceae counts were quantified in violet red bile dextrose agar (VRBDA, Merck) after incubation at 37 °C for 24 h by counting the red-pink colonies formed [18]. Total coliforms and *Escherichia coli* counts were enumerated in Chromocult® Coliform Agar (CCA, Merck) after incubation at 37 °C for 24 h, with pink colonies being classified as total coliforms, whereas dark-blue colonies were classified as presumptive *E. coli* colonies [18]. The results were expressed in all cases as the logarithm of colony-forming units per milliliter (log CFU/mL), with all samples analyzed in triplicate.

### 2.3. Immunoglobulin Content and Lysozyme Activity Analyses

IgG, IgM and IgA were measured in donkey colostrum using specific human ELISA kits (KOMA BIOTECH, Seul, Republic of Korea), according to the manufacturer's instructions (reference K3231067, K0231069 and K0231064, respectively). ELISA (or enzyme-linked immunosorbent assay) is an immunoassay technique involving the reaction of antigens and antibodies in vitro. The kits contained all of the required reagents and material for the specific Ig quantitation: a precoated 96-well ELISA microplate (with antigen-affinity purified goat anti-human Ig), plate sealers, detection antibody (horseradish peroxidase-conjugated antigen-affinity purified goat anti-human Ig), standard protein (human reference serum), assay diluent (1% bovine serum albumin), color development reagents (tetramethylbenzidine and $H_2O_2$ solutions), stop solution (2 M $H_2SO_4$) and washing solution (phosphate-buffered saline powder with 0.05% Tween-20, pH 7.4). To each well of the precoated microplate, 100 µL of blank, standard or donkey colostrum sample was added and incubated at room temperature for 1 h with the plate sealer provided. The standards were diluted following the manufacturer's recommended dilutions. Then, 100 µL of the diluted detection antibody (1:20,000 to 1:50,000) was added per well and incubated for 1 h at room temperature in the sealed plate. Before, between and after the above steps, the wells were aspirated to remove the liquid and washed five times with the washing solution to remove unbound molecules. For the color development reaction, 100 µL of color development solution was added to the wells and incubated for about 10 min. The reaction was stopped by adding 100 µL of the stop solution, and the absorbance was read at 450 nm in less than 20 min using a Multiskan GO microplate Spectrophotometer (Thermo Scientific, Waltham, MA, USA). Triplicate determinations were performed for each Ig, and total concentrations were determined using the standard curves constructed with the diluted standards.

Lysozyme activity was determined using a *Micrococcus lysodeikticus*-based turbidimetric procedure recommended by Sigma Chemical Co. (Catalog Number MAK393, Sigma, St. Louis, MO, USA), with slight modifications, with one unit of enzyme being defined as the amount of enzyme that causes a decrement of 0.001 units of absorbency at 450 nm. The principle of this assay is the lytic activity of lysozyme towards *Micrococcus lysodeikticus* cell walls (antibacterial activity) by measuring the loss of light intensity in the direction of incident beam propagation, with reference to a standard solution. A 0.015% (*w/v*) *Micrococcus lysodeikticus* cell suspension (substrate) was prepared by suspending *Micrococcus lysodeikticus* ATCC 4698 lyophilized cells in sodium phosphate buffer (66 mM, pH 6.24). To assure the freshness of this suspension, its absorbance at 450 nm (A450 nm) was measured at the beginning of the analyses and was consistently between 0.6 and 0.7, as it should be. Prior to the enzymatic reaction, the substrate was heated at 30 °C in a thermostatic bath. The reaction was initiated by adding 0.10 mL of appropriately diluted donkey colostrum to 2.5 mL of substrate and, after mixing by inversion, the decrease at 450 nm was immediately recorded for 3 min at 10 s intervals using a PerkinElmer Lambda 35 UV–Vis spectrophotometer (PerkinElmer Instruments, Waltham, MA, USA). Measurements were carried out against a blank containing the substrate and 0.10 mL of buffer, and each sample was measured in triplicate.

For Ig content and lysozyme activity, the results were expressed as percentages (%), considering 100% the value obtained for colostrum before any treatment (untreated colostrum).

*2.4. Kinetic Data Analysis*

Decrements in IgG at 550 and 625 MPa and IgM at 550 MPa could be described by a first-order model (in the other studied conditions, the number of data points with measurable Ig content was not enough to allow kinetic analysis). According to Equation (1), the Ig concentration loss rate ($-dA/dt$) is proportional to the inactivation rate constant ($k$) and the Ig concentration at each treatment time (A).

$$\frac{dA}{dt} = -kA \tag{1}$$

The reaction rate constant was determined from a semi-logarithmic plot (Equation (2)) of the relative Ig retention ($A/A_0$) as a function of the exposure time (t), with A representing the value after HPP treatment and $A_0$ the initial value.

$$\ln\left(\frac{A}{A_0}\right) = -kt \tag{2}$$

D-values (decimal reduction time, the time needed to observe a decrease of 90% of the initial value) were also calculated according to Equation (3).

$$D = \frac{\ln(10)}{k} \tag{3}$$

*2.5. Statistical Analysis*

The effects of the applied treatments were tested in a one-way analysis of variance (ANOVA), followed by a multiple comparisons test (Tukey's HSD) to find which treatments were significantly different from one another at a 0.05 level of significance.

## 3. Results and Discussion

*3.1. Microbial Load Analysis*

As displayed in Table 1, the initial TAM and Enterobacteriaceae counts in untreated donkey colostrum (day 0) were below the detection limit (<1.0 log CFU/mL), and during refrigeration, there was an increase in TAM and Enterobacteriaceae counts in untreated donkey colostrum at day 4 (6.7 log CFU/mL) and day 7 (3.5 log CFU/mL), respectively, surpassing 8.0 log CFU/mL after 18 days for both microorganisms. Coliforms as well as presumptive *E. coli* (the most common contaminant of untreated and processed milk and dairy products [19]) were also not detected in raw donkey colostrum. The low microbial counts in raw donkey colostrum found in this work indicate good hygienic conditions during the collection of the colostrum and are also possibly due to the presence of natural antimicrobial substances, such as lysozyme and Igs [17], since it is reported in the literature that these substances may work synergistically in order to inhibit bacterial growth. For instance, Zhang et al. (2008) [17] did not verify microflora variations in donkey milk after 96 h of storage at 4 °C except for coliforms, which showed a 1.0 log CFU/mL increase. In the literature, nevertheless, the results from a biennial study on untreated donkey milk showed an occasional isolation of *E. coli* and a wide variability in coliform contamination (<1.0–6.6 log CFU/mL; [19]), which indicates environmental sources of contamination [20].

After TP and HPP treatments and subsequent refrigerated storage for 40 days, all microorganisms studied remained below the detection limit (<1.0 log CFU/mL), pointing to both treatment processes assuring an effective pasteurization effect (Table 1).

**Table 1.** Microbial counts in untreated donkey colostrum and after thermal (TP) and high-pressure pasteurization (HPP). Data are presented as log CFU/mL.

| Treatment | | Storage Days | Total Aerobic Mesophilic | Enterobacteriaceae | Total Coliforms |
|---|---|---|---|---|---|
| **Untreated** | | 0 | <1.0 | <1.0 | |
| | | 4 | 6.7 ± 0.1 | <1.0 | |
| | | 7 | 5.1 ± 0.1 | 3.5 ± 0.1 | <1.0 |
| | | 11 | 5.3 ± 1.9 | 4.4 ± 0.1 | |
| | | 18 | 8.0 ± 2.5 | 8.2 ± 0.1 | |
| **TP** | | 0 | | | |
| | | 4 | | | |
| | | 7 | | | |
| | | 11 | | | |
| | | 18 | <1.0 | <1.0 | <1.0 |
| | | 21 | | | |
| | | 28 | | | |
| | | 40 | | | |
| **HPP** | **400 MPa** | 0 | | | |
| | | 4 | | | |
| | | 7 | | | |
| | | 11 | | | |
| | | 18 | <1.0 | <1.0 | <1.0 |
| | | 21 | | | |
| | | 28 | | | |
| | | 40 | | | |
| | **550 MPa** | 0 | | | |
| | | 4 | | | |
| | | 7 | | | |
| | | 11 | | | |
| | | 18 | <1.0 | <1.0 | <1.0 |
| | | 21 | | | |
| | | 28 | | | |
| | | 40 | | | |

### 3.2. Immnunoglobulin Content and Lysozyme Activity in Raw Colostrum

According to our results, IgM represents 86% of the total Igs studied, followed by IgA and IgG with 8 and 6%, respectively, showing a different relative profile compared to human colostrum [21], where IgA was reported as the primary Ig [7], with values that can reach 8.0–9.0 mg/mL [22,23]. Moreover, it has been widely described in the literature that the Ig content in colostrum is highly dependent on several factors, such as the animal species, the stage of lactation and the time of collection, among others [24]. Since the equine placenta does not allow any antibody transfer, the foal immunity depends entirely on colostrum and milk intake in the first days post-partum [25].

The enzyme lysozyme, together with other factors including Igs, is one of the most important natural antimicrobial compounds in colostrum and milk, being present in donkey milk in high quantities [26]. In the present study, the lysozyme activity found in donkey colostrum was $3.54 \pm 0.87$ ($\times 10^5$) U/mL, which is very similar to previously reported

values for donkey colostrum, 1.4–1.2 ($\times 10^5$) U/mL, at day one of lactation [27], while for donkey milk, the same authors reported a value of 3.4–2.0 ($\times 10^4$) U/mL at day 60 of lactation, indicating that the amount of lysozyme varies considerably along lactation. These results are very interesting since the higher lysozyme content in donkey colostrum is an advantage to the safety and functionality of the donkey colostrum for increased periods.

### 3.3. Effect of Thermal and High-Pressure Pasteurization

3.3.1. Immunoglobulins

- IgG

As can be seen in Figure 1a, TP caused an approximately 79% decrease in the IgG content compared to the untreated samples. In the literature, information regarding the effect of temperature on IgG present in colostrum is very scarce, but in human colostrum, Koenig et al. (2005) [28] reported a similar reduction ($\approx$72%), while Sousa et al. (2014) [7] observed only a 23% reduction in IgG caused by Holder pasteurization. These decrements are lower than those verified for purified bovine milk IgG in other studies [29,30] pasteurized by TP, suggesting that several colostrum components, such as fat, casein, salts and sugars, such as lactose, may protect IgG during TP [31].

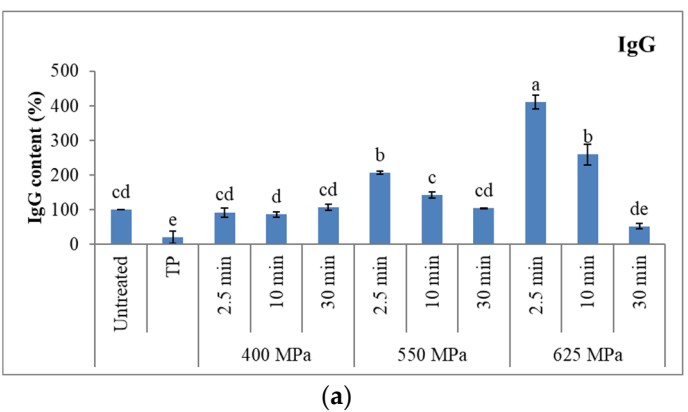
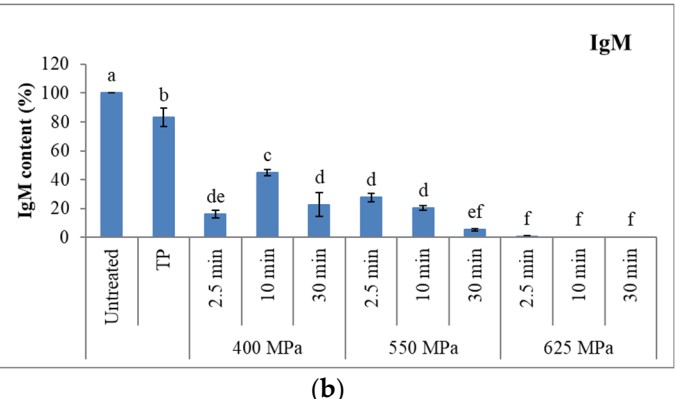

(**a**)  (**b**)

**Figure 1.** IgG (**a**) and IgM (**b**) content (%) in untreated donkey colostrum and after thermal (TP) and high-pressure pasteurization (HPP). Data are presented as a percentage considering 100% as the value obtained for colostrum before any treatment (untreated colostrum). Different letters denote significant differences ($p < 0.05$) between the different conditions (a–f).

Regarding HPP, the IgG content at 400 MPa was preserved ($p > 0.05$) (Figure 1a), similar to results in the literature for human colostrum treated at low and mild pressures (200 and 400 MPa) [7]. On the other hand, for the pressure/holding times conditions of 550 MPa/2.5 min, 625 MPa/2.5 and 10 min, a significant increase ($p < 0.05$) in the IgG content was observed, reaching a maximum of about 4-fold for 625 MPa/2.5 min and about 2.5-fold for 625 MPa/10 min. The reason for this can be due to the release of IgG that might be bound to other components, such as lactoferrin [32]. On the contrary, an increase in the holding time to 30 min, at 625 MPa, caused a reduction in the IgG content to about 50% compared to untreated colostrum, possibly due to inactivation of IgG after release from other components for longer pressure holding times [33]. Foster et al. (2016) [12] verified that the IgG content in bovine colostrum decreased by about 5% at 400 MPa with the increase in the holding time from 10 to 20 min, but no significant differences were obtained at 300 MPa.

- IgM

TP and HPP caused a decrease ($p < 0.05$) in IgM content (Figure 1b) but to a much lower extent in the case of TP, resulting in a reduction of 17%, while HPP caused decrements from 55% (400 MPa/10 min) to below the detection limit (in this case, particularly for 625 MPa). IgM seemed to be more heat stable than IgG, similar to the findings reported by

Sousa et al. (2014) [7] for human colostrum. At the same time, IgM in donkey colostrum seemed to be more pressure sensitive than in human colostrum for which no reduction effect at 400/30 min was verified, and an IgM retention of about 40% at 600 MPa/30 min was discovered [7].

- IgA

IgA content in donkey colostrum was found to be reduced to levels below the detection limit after TP and HPP. Sousa et al. (2014) [7] observed IgA retention on human colostrum for several pressure treatments ranging from 200 to 600 MPa (with holding times between 2.5 and 30 min), with a 74% retention even at 600 MPa (30 min, 8 °C), similar to that observed after TP. In the present study, and compared to IgG and IgM, IgA could be considered both more heat and pressure sensitive in human colostrum.

- Kinetics analysis for immunoglobulins

The inactivation kinetic parameters $k$ (reaction rate constant, $min^{-1}$) and D-values (min) were calculated for cases where enough data were available and are presented in Table 2. To the authors' best knowledge, there is no work reporting $k$ and D-values for Igs losses caused by TP and HPP treatments in donkey colostrum. The D-value of IgG for 550 MPa was 2.6-fold higher than IgM, thus confirming that, at this pressure, IgG is more pressure resistant than IgM in donkey colostrum, while for IgG, the D-value at 550 MPa is about 3.3-fold higher than the one at 625 MPa. For human colostrum, Sousa et al. (2014) [7] reported 235 and 40 min as the D-values for IgG and IgM at 600 MPa, respectively, which indicates that, as in the present work, IgG is also more pressure stable than IgM.

**Table 2.** HPP inactivation kinetic parameters of IgM and IgG in donkey colostrum.

| Protein | Pressure (MPa) | $k$ ($min^{-1}$) | $R^2$ | D-Value (min) |
|---------|----------------|------------------|-------|---------------|
| IgG | 550 | $2.31 \times 10^{-2} \pm 7.93 \times 10^{-3}$ | 0.910 | $99.68 \pm 18.01$ |
| | 625 | $7.57 \times 10^{-2} \pm 3.83 \times 10^{-3}$ | 0.997 | $30.42 \pm 1.57$ |
| IgM | 550 | $6.05 \times 10^{-2} \pm 5.60 \times 10^{-3}$ | 0.992 | $38.06 \pm 3.56$ |

### 3.3.2. Lysozyme Activity

The percentage of lysozyme activity retention found in donkey colostrum after TP and HPP is presented in Figure 2, with the results showing a significant decrease ($p < 0.05$) in lysozyme activity of 37% after TP treatment, a lower reduction compared to that reported for human colostrum (44 to 85%) using the same TP conditions [7,28]. For HPP, using pressure conditions of 400 and 550 MPa, losses from 25 to 34% of lysozyme activity were observed. For the highest pressure/holding time conditions of 625 MPa for 10 and 30 min, a reduction of about 42% was observed, which was not statistically different ($p > 0.05$) from TP treatment. In human colostrum, changes to lysozyme activity at pressure levels of 200–600 MPa/2.5–30 min were not observed [7]. Thus, HPP preserves equal to better lysozyme activity compared to TP.

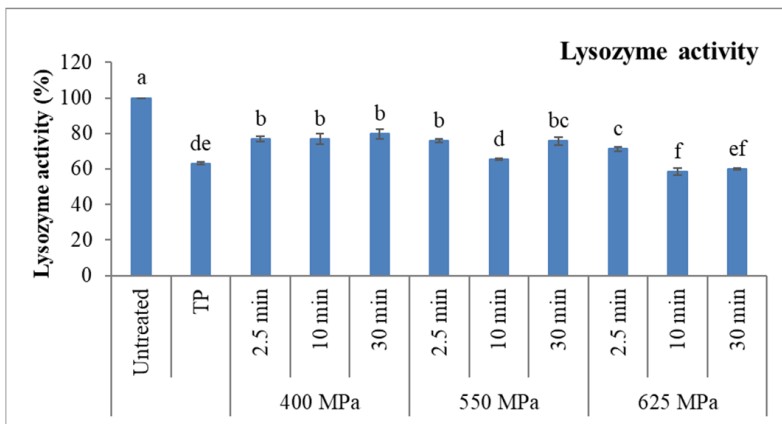

**Figure 2.** Lysozyme activity (%) in untreated donkey colostrum and after thermal (TP) and high-pressure pasteurization (HPP). Data are presented as a percentage considering 100% as the value obtained for raw colostrum. Different letters denote significant differences ($p < 0.05$) between the different conditions (a–f).

## 4. Conclusions

This study meticulously examined the impact of thermal pasteurization (TP) and high-pressure pasteurization (HPP) treatments on the microbial quality, immunoglobulin (IgA, IgM and IgG) content and lysozyme activity of donkey colostrum. The results indicate that HPP at 550 MPa (2.5 min) ensures microbial safety while preserving lysozyme activity and IgG content, presenting a viable alternative to TP. Notably, TP resulted in a significant reduction in IgG content, contrasting with HPP, which demonstrated negligible effects on immunoglobulin content. Furthermore, TP exhibited superior efficacy in retaining IgM values compared to HPP. These findings contribute valuable insights into the unique characteristics of donkey colostrum, crucial for devising effective processing techniques to preserve its nutritional and therapeutic properties.

**Author Contributions:** Conceptualization, J.A.S. and S.G.S.; data curation, M.S.G., S.G.S. and L.G.F.; formal analysis, L.G.F. and M.S.G.; funding acquisition, J.A.S.; investigation, M.S.G., S.G.S., L.G.F. and R.P.Q.; methodology, M.S.G., S.G.S. and J.A.S.; project administration, J.A.S.; resources, J.A.S.; supervision, J.A.S.; validation, S.G.S., L.G.F. and J.A.S.; visualization, M.S.G., S.M.C., L.G.F. and C.A.P.; writing—original draft preparation, M.S.G., S.M.C. and L.G.F.; writing—review and editing, J.A.S., L.G.F. and C.A.P. All authors have read and agreed to the published version of the manuscript.

**Funding:** This work received financial support from FCT/MCTES (LA/P/0008/2020 DOI 10.54499/ LA/P/0008/2020, UIDP/50006/2020 DOI 10.54499/UIDP/50006/2020 and UIDB/50006/2020 DOI 10.54499/UIDB/50006/2020) through national funds.

**Institutional Review Board Statement:** Not applicable.

**Informed Consent Statement:** Not applicable.

**Data Availability Statement:** Data are available upon request to the corresponding author.

**Acknowledgments:** This work received support and help from FCT/MCTES (LA/P/0008/2020 DOI 10.54499/LA/P/0008/2020, UIDP/50006/2020 DOI 10.54499/UIDP/50006/2020 and UIDB/50006/ 2020 DOI 10.54499/UIDB/50006/2020), through national funds.

**Conflicts of Interest:** The author, Rui P. Queirós, was employed by the company Hiperbaric SA. The remaining authors declare that the research was conducted in the absence of any commercial or financial relationships that could be construed as a potential conflict of interest.

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
