# Peer review of "Comparison of Thermal and High-Pressure Pasteurization on Immunoglobulins, Lysozyme and Microbial Quality of Donkey Colostrum"

_applsci, doi:10.3390/app14041592_

Round 1

Reviewer 1 Report

Comments and Suggestions for Authors

The current study seeks to compare the effects of thermal (TP) and high-pressure processing (HPP) on some quality attributes of donkey colostrum. Overall, the results show that HPP treatments can provide good results in maintaining product quality compared to TP. The results are clearly presented, but in the conclusions section, the authors could explicitly point out the advantages and disadvantages of the HPP in comparison to TP in a fair way. For instance, TP was more effective in keeping the IgM values. In addition, in Table 1, please provide the standard errors associated with the rate constants estimates. 

Comments on the Quality of English Language

There are some minor mistakes in the manuscript. For instance, replace studied with study in line 61. Please revise the whole manuscript to find additional errors.

Author Response

Comments and Suggestions for Authors

The current study seeks to compare the effects of thermal (TP) and high-pressure processing (HPP) on some quality attributes of donkey colostrum. Overall, the results show that HPP treatments can provide good results in maintaining product quality compared to TP. The results are clearly presented, but in the conclusions section, the authors could explicitly point out the advantages and disadvantages of the HPP in comparison to TP in a fair way. For instance, TP was more effective in keeping the IgM values. In addition, in Table 1, please provide the standard errors associated with the rate constants estimates. 

R: Thank you for your valuable feedback on our study. The conclusions section was revised and completed accordantly. Concerning table 1, standard errors were also added.

Comments on the Quality of English Language

There are some minor mistakes in the manuscript. For instance, replace studied with study in line 61. Please revise the whole manuscript to find additional errors.

R: The manuscript was reviewed and corrected accordingly using “track changes”.

Reviewer 2 Report

Comments and Suggestions for Authors

Among the currently used methods of preserving donor breast milk, the most common is thermal treatment (pasteurization). This method of preservation reduces the level of microorganisms, but may cause the degradation of valuable ingredients, such as: antibodies, enzymes and vitamins. High pressure processing (HPP) (also referred to as high hydrostatic pressure, HHP) is a non-thermal pasteurization technology that relies on very high pressures (400–600 MPa) to inactivate pathogens. Instead of heat, thus causing less negative impact in the food nutrients and quality. The goal in the field is therefore to use a processing technology that, in addition to microbiological safety, would preserve the natural or create new, beneficial quality features of the product, such as bioactivity or durability, while minimizing undesirable changes caused by processing. One of the techniques used is high pressure treatment, called pressure treatment, pascalization, high pressure treatment or pressurization. The high hydrostatic pressure (HPP) technique, which is a modern technology for preserving and preserving the sensory properties of food, can be an alternative to thermal processing of many food products, as well as breast milk. The authors used this modern HPP method for research on donkey colostrum more valuable early milk. It is known that donkey milk is most similar to human milk in terms of the structure of its protein fraction, as it has a low casein content and a relatively high whey protein content. This experimental and promising research in the Spanish-Portuguese cooperation is very interesting and deserves publication. the colostrum was equally divided for thermal Holter method (TP) and high-pressure pasteurization (HPP).  The study is based on measurement of activity analysis immunoglobulins IgG, IgA i IgM, lysozyme and microbial quality.The methodology is clearly and detailed described. The microbiological analysis of the quality of colostrum included measurement of Escherichia coli and Entereobactariacae.  It is worth emphasizing that all samples were analyzed in triplicate. This makes the results more precise. Interesting results concern immunoglobulins and lysozyme. Proving the   activity of lysozyme in later days, even on the 60th day of lactation, is extremely important for long-term anti-infection effectiveness. This study also showed that HPP is much better method to preserve IgG content. This original and concise paper is interesting research  that constitutes an important contribution to the search for the most effective methods of preserving the activity of  individual of components of donor milk. Do the authors think that donkey can be used in the human milk banks and for  of human milk fortification. The conclusions are specific, short, well-develop The list of references contains 32 well-selected items. However, that the authors do not cited Polish paper by researchers whose device (Uni- 119press Equipment, Model U33, Warsaw, Poland), was used in the research methodology. The figures are clear and the legends below them are comprehensive. This is an interesting original work that fully deserves publication.

Author Response

Comments and Suggestions for Authors

Among the currently used methods of preserving donor breast milk, the most common is thermal treatment (pasteurization). This method of preservation reduces the level of microorganisms, but may cause the degradation of valuable ingredients, such as: antibodies, enzymes and vitamins. High pressure processing (HPP) (also referred to as high hydrostatic pressure, HHP) is a non-thermal pasteurization technology that relies on very high pressures (400–600 MPa) to inactivate pathogens. Instead of heat, thus causing less negative impact in the food nutrients and quality. The goal in the field is therefore to use a processing technology that, in addition to microbiological safety, would preserve the natural or create new, beneficial quality features of the product, such as bioactivity or durability, while minimizing undesirable changes caused by processing. One of the techniques used is high pressure treatment, called pressure treatment, pascalization, high pressure treatment or pressurization. The high hydrostatic pressure (HPP) technique, which is a modern technology for preserving and preserving the sensory properties of food, can be an alternative to thermal processing of many food products, as well as breast milk. The authors used this modern HPP method for research on donkey colostrum more valuable early milk. It is known that donkey milk is most similar to human milk in terms of the structure of its protein fraction, as it has a low casein content and a relatively high whey protein content. This experimental and promising research in the Spanish-Portuguese cooperation is very interesting and deserves publication. The colostrum was equally divided for thermal Holter method (TP) and high-pressure pasteurization (HPP).  The study is based on measurement of activity analysis immunoglobulins IgG, IgA i IgM, lysozyme and microbial quality. The methodology is clearly and detailed described. The microbiological analysis of the quality of colostrum included measurement of Escherichia coli and Entereobactariacae.  It is worth emphasizing that all samples were analysed in triplicate. This makes the results more precise. Interesting results concern immunoglobulins and lysozyme. Proving the   activity of lysozyme in later days, even on the 60th day of lactation, is extremely important for long-term anti-infection effectiveness. This study also showed that HPP is much better method to preserve IgG content. This original and concise paper is interesting research that constitutes an important contribution to the search for the most effective methods of preserving the activity of individual of components of donor milk. Do the authors think that donkey can be used in the human milk banks and for of human milk fortification. The conclusions are specific, short, well-develop. The list of references contains 32 well-selected items. However, that the authors do not cited Polish paper by researchers whose device (Uni- 119press Equipment, Model U33, Warsaw, Poland), was used in the research methodology. The figures are clear and the legends below them are comprehensive. This is an interesting original work that fully deserves publication.

R: Thank you for your very positive feedback on our work. The authors proceeded to properly cite the recommended paper. Now it reads, in the introduction section: “In another study, Wesolowska and colleagues (2018) [14] compared the effects of single and cycled HPP  treatments (600 MPa/10 min, 100 MPa/10 min - 10 min interval - 600 MPa/10 min, 200 MPa/10 min – 10 min interval – 400 MPa/10 min, and 200 MPa/10 min – 10 min interval – 600 MPa/10 min, at 19-21 °C) and traditional holder pasteurization (62.5 °C, 30 min) in human milk. The results showed that the condition 200 + 400 MPa was able to preserve better the IgG levels compared to traditional hold-er pasteurization (82.24 versus 50.96%, respectively), as well as the levels of hepatocyte growth factor (97.15 versus 11.28%, respectively). The authors also reported that the content of leptin increased for all HPP conditions tested compared to holder pasteurization, as well as a better retention of insulin.

Reviewer 3 Report

Comments and Suggestions for Authors

The paper titled "Comparison of Thermal and High-Pressure Pasteurization on Immunoglobulins, Lysozyme, and Microbial Quality of Donkey Colostrum" provides valuable insights into the effects of thermal pasteurization (TP) and high-pressure pasteurization (HPP) on the composition and quality of donkey colostrum. The study is commendable for its systematic approach and thorough evaluation of various parameters.

The authors rigorously examined the impact of TP and HPP with three on immunoglobulins (IgG, IgM, and IgA) content, lysozyme activity, and microbial load in donkey colostrum stored at 4 °C for 40 days. The results indicate intriguing variations in the levels of these components under different pasteurization conditions.

One noteworthy finding is that the IgG levels with HPP at 400 MPa did not considerably affect the immunoglobin content, contrasting with a 90% decrease observed with TP.

The study also delves into lysozyme activity, revealing a higher overall decrease after TP compared to HPP. This nuanced examination provides a comprehensive understanding of how different pasteurization techniques affect the vital components of donkey colostrum.

Furthermore, the authors assessed microbial quality, and both total aerobic mesophiles and Enterobacteriaceae remained below detection limits after 40 days of refrigerated storage for both TP and HPP. This suggests that both methods effectively preserve the microbial quality of donkey colostrum.

The paper concludes by suggesting that HPP could be a potential alternative to conventional TP for preserving donkey colostrum, particularly due to its favorable retention of IgG and lysozyme activity. The acknowledgment that this is the first study evaluating the effects of HPP on donkey colostrum adds significance to the research.

In summary, this paper significantly contributes to understanding pasteurization techniques on donkey colostrum quality. The detailed analysis and conclusive findings pave the way for further research in this field. Researchers and food science and nutrition professionals will find this paper a valuable resource for exploring alternative preservation methods for colostrum.

I appreciate the thorough investigation presented in your paper titled "Comparison of Thermal and High-Pressure Pasteurization on Immunoglobulins, Lysozyme, and Microbial Quality of Donkey Colostrum." This study sheds light on the effects of different pasteurization techniques on key components of donkey colostrum.

While the research is commendable, I want to raise a concern regarding the limited sample size mentioned in the methodology. The colostrum was collected from a single apparently healthy donkey female one day after partum. It was then transported to the laboratory within four hours under refrigerated conditions (4 °C). The use of a single donkey for the study raises questions about the generalizability of the findings.

Considering the biological variability that may exist among different individuals, it would be beneficial to elaborate on the rationale behind choosing a single donor for the study. Additionally, discussing the potential impact of individual variation on the study outcomes would provide readers with a more comprehensive understanding of the limitations associated with the sample size.

Expanding the discussion on this aspect would strengthen the robustness of your research and enhance the credibility of the conclusions drawn from the study. Addressing this concern would contribute to the overall quality and impact of your valuable research.

Comments on the Quality of English Language

Overall, the language is fine.

Author Response

Comments and Suggestions for Authors

The paper titled "Comparison of Thermal and High-Pressure Pasteurization on Immunoglobulins, Lysozyme, and Microbial Quality of Donkey Colostrum" provides valuable insights into the effects of thermal pasteurization (TP) and high-pressure pasteurization (HPP) on the composition and quality of donkey colostrum. The study is commendable for its systematic approach and thorough evaluation of various parameters. The authors rigorously examined the impact of TP and HPP with three on immunoglobulins (IgG, IgM, and IgA) content, lysozyme activity, and microbial load in donkey colostrum stored at 4 °C for 40 days. The results indicate intriguing variations in the levels of these components under different pasteurization conditions. One noteworthy finding is that the IgG levels with HPP at 400 MPa did not considerably affect the immunoglobin content, contrasting with a 90% decrease observed with TP.  The study also delves into lysozyme activity, revealing a higher overall decrease after TP compared to HPP. This nuanced examination provides a comprehensive understanding of how different pasteurization techniques affect the vital components of donkey colostrum. Furthermore, the authors assessed microbial quality, and both total aerobic mesophiles and Enterobacteriaceae remained below detection limits after 40 days of refrigerated storage for both TP and HPP. This suggests that both methods effectively preserve the microbial quality of donkey colostrum. The paper concludes by suggesting that HPP could be a potential alternative to conventional TP for preserving donkey colostrum, particularly due to its favorable retention of IgG and lysozyme activity. The acknowledgment that this is the first study evaluating the effects of HPP on donkey colostrum adds significance to the research. In summary, this paper significantly contributes to understanding pasteurization techniques on donkey colostrum quality. The detailed analysis and conclusive findings pave the way for further research in this field. Researchers and food science and nutrition professionals will find this paper a valuable resource for exploring alternative preservation methods for colostrum. I appreciate the thorough investigation presented in your paper titled "Comparison of Thermal and High-Pressure Pasteurization on Immunoglobulins, Lysozyme, and Microbial Quality of Donkey Colostrum." This study sheds light on the effects of different pasteurization techniques on key components of donkey colostrum.

R:  Thank you for your thoughtful review and positive feedback. We appreciate your recognition of our study. We are grateful for the acknowledgment of our paper's contribution to the understanding of the possibility to apply high pressure processing to preserve donkey colostrum quality.

While the research is commendable, I want to raise a concern regarding the limited sample size mentioned in the methodology. The colostrum was collected from a single apparently healthy donkey female one day after partum. It was then transported to the laboratory within four hours under refrigerated conditions (4 °C). The use of a single donkey for the study raises questions about the generalizability of the findings. Considering the biological variability that may exist among different individuals, it would be beneficial to elaborate on the rationale behind choosing a single donor for the study. Additionally, discussing the potential impact of individual variation on the study outcomes would provide readers with a more comprehensive understanding of the limitations associated with the sample size. Expanding the discussion on this aspect would strengthen the robustness of your research and enhance the credibility of the conclusions drawn from the study. Addressing this concern would contribute to the overall quality and impact of your valuable research.

R: Thank you for your thoughtful and constructive feedback on our research. We appreciate your diligence in examining the methodology and raising concerns about the limited sample size, which we agree. At the time of this study, the farm had only one female with available colostrum suitable for our research. The limited availability of colostrum for research purposes was due to its critical role in supporting the health of baby donkeys on the farm, and the donation volume was restricted accordingly. We acknowledge the significance of providing a more detailed explanation for the choice of a single donor and the associated constraints.

Now we can read: “The colostrum was collected from an apparently healthy donkey female, one day after partum and transported to the laboratory within four hours in refrigerated conditions (4 °C). At the time of this study, the farm had only one female with available colostrum suitable for our research. The limited availability of colostrum for research purposes was due to its critical role in supporting the health of foals on the farm, and the donation volume was restricted accordingly.”